# Feature Fusion Based on Temporal-Spatial Attention Model for Automatic Epileptic Seizure Detection

Xiang Li[1#]
*School of Medicine*
*The Chinese University of Hong Kong, Shenzhen*
Shenzhen, China
224050069@link.cuhk.edu.cn

Zhiheng Zhang[2#]
*School of Data Science*
*The Chinese University of Hong Kong, Shenzhen*
Shenzhen, China
12309080@link.cuhk.edu.cn

Zhaonian Guo[3]
*School of Sicence and Engineering*
*The Chinese University of Hong Kong, Shenzhen*
Shenzhen, China
122090144@link.cuhk.edu.cn

Xin Wang[4]
*Research Center for Neural Engineering*
*Shenzhen Institutes of Advanced Technology*
Shenzhen, China
wangxin@siat.ac.cn

Ke Zhang[5]
*School of Medicine*
*The Chinese University of Hong Kong, Shenzhen*
Shenzhen, China
kezhang@cug.edu.cn

Shixiong Chen*
*School of Medicine*
*The Chinese University of Hong Kong, Shenzhen*
Shenzhen, China
chenshixiong@cuhk.edu.cn

*Abstract*—Epilepsy, a widespread neurological disorder, poses challenges for timely seizure detection due to complex EEG signals, noise, and data imbalance. We propose the Temporal-Spatial Fusion Attention (TSFA) model, which integrates Bi-LSTM for temporal feature extraction and Pearson correlation for spatial dependencies, fused via an attention mechanism to enhance seizure detection. Evaluated on the CHB-MIT dataset, TSFA achieves 97.15% accuracy, 97.06% sensitivity, and 96.45% F1-score, surpassing existing methods. Ablation studies highlight the synergy of temporal and spatial attention, ensuring robustness and generalization. TSFA shows promise for clinical applications, with future work targeting multimodal learning.

*Index Terms*—Epileptic seizure detection, EEG, Temporal-Spatial Attention, Deep Learning, Feature Fusion.

## I. INTRODUCTION

Epilepsy is a common neurological disorder characterized by repeated episodes of abnormal electrical activity in the brain, often accompanied by sudden loss of consciousness, muscle spasms, or involuntary behavior [1]. Globally, the prevalence of epilepsy is high, affecting the lives of millions of people. According to the World Health Organization (WHO), about 5% of people have experienced at least one seizure in their lives. There are various types of seizures, including focal seizures, generalized seizures, and non-convulsive seizures, and the symptoms and manifestations vary from individual to individual. Due to the abruptness and unpredictability of epilepsy, it has a serious impact on the quality of life, work ability, and social function of patients, so timely and effective seizure detection and early warning has become one of the key problems in clinical treatment [2]. In recent years, with the rapid development of deep learning technology, the use of electroencephalography (EEG) signals for seizure detection has become a hot research direction. EEG can capture subtle changes in the brain's electrical activity, so it is of great significance in the diagnosis and early warning of epilepsy. However, because of the complexity of the seizure signal and the high noise problem, how to efficiently and accurately identify the seizure from the EEG data is still a challenging task. Deep learning models, especially convolutional neural networks (CNN) [3], long and short-term memory networks (LSTM) [4], and graph neural networks (GNN) [5] , have been widely used in the field of automatic detection and prediction of epileptic seizures. Shen et al. (2022) [6] developed a robust EEG-based seizure detection method using a discrete wavelet transform and a support vector machine classifier, while Huang et al. [7] enhanced detection by incorporating a dual attention mechanism to capture subtle epileptic patterns. Based on the above problems, we propose a Spatio-Temporal method, the Temporal-Spatial Fusion Attention (TSFA) model, which takes into account both temporal and spatial information by introducing spatio-temporal Attention so as to improve the

XiangLi and Zhiheng Zhang contributed equally to this work.
*Corresponding author: Shixiong Chen.
This work was supported in part by the National Key RD Program of China (2022YFE0197500), National Natural Science Foundation of China (62471422, 62401558), Shenzhen Medical Research Fund (D2402003), the Science and Technology Innovation Special Fund for Medical Health Technology Research and Development of Longgang District, Shenzhen (LGKCYLWS2024-16), and the Science and Technology Planning Project of Shenzhen(JCYJ20230807093819039).

detection accuracy of epileptic seizures. Traditional methods usually focus only on features in time or space dimensions, whereas seizures often involve complex changes in EEG signals in both time and space dimensions. Therefore, our model is able to automatically weight temporal and spatial features through the spatio-temporal attention mechanism, fusing the associations between different brain regions to capture the underlying pattern of seizures. Based on the CHB-MIT dataset, we obtained an accuracy of 97.15% on epileptic seizure detection. This spatio-temporal fusion method demonstrates superior robustness and high detection performance for epileptic seizure detection.

## II. METHOD

The complete model contains three parts, as illustrated in Fig. 1. Our model integrates temporal and spatial attention mechanisms. Temporal features are extracted using Bi-LSTM, while spatial features are captured through Pearson correlation calculated from segmented time windows. These features serve as inputs to the Temporal and Spatial Attention modules. Finally, the outputs are fused through a Fusion Attention mechanism to enhance feature representation for seizure detection.

### A. TSFA: Temporal-Spatial Fusion Attention

To enhance the feature capture performance of EEG signals, we design the Temporal-Spatial Fusion Attention (TSFA) model, which combines the Temporal Attention and Spatial Attention to effectively capture both dynamic temporal patterns and inter-channel spatial dependencies. The purpose of this model is to refine the temporal features extracted from Bi-LSTM while emphasizing the most informative channels, ensuring that the model focuses more effectively on discriminative EEG representations for seizure detection.

#### 1) Temporal Feature

This research focuses on extracting the temporal features of EEG signals. Given the strong ability of Bidirectional Long Short-Term Memory (Bi-LSTM) [8] networks to capture sequential dependencies, we employ Bi-LSTM to model the temporal relationships between sliding time windows. To achieve this, EEG signals are segmented into 1-second time windows, ensuring that each window contains a fixed length of signal data. Formally, each window is represented as $W \in \mathbb{C}^{C \times T}$, where $C$ denotes the number of EEG channels and $T$ represents the number of temporal steps. The forward and backward hidden states of the Bi-LSTM are computed as follows:

$$fh_t = \text{LSTM}_{\text{forward}}(x_t, fh_{t-1}) \tag{1}$$

$$bh_t = \text{LSTM}_{\text{backward}}(x_t, bh_{t-1}) \tag{2}$$

In the formulas, $t \in [1, T]$. The final temporal feature is obtained by joining the forward and backward hidden states as follows:

$$h_t = [fh_t; bh_t]$$

The temporal attention part is conducted based on a fixed-size $3 \times 3$ initial window, which slides on the EEG sequence and computes the local features. For every temporal step $t$,

there are different weights and, consequently, different $Q$, $K$, $V$ matrices. Input temporal features were processed by Local Attention Mechanism to derive Query ($Q$), Key ($K$), and Value ($V$) matrices, using the following formulas:

$$Q_t = W_q F^w \tag{4}$$

$$K_t = W_k F^w \tag{5}$$

$$V_t = W_v F^w \tag{6}$$

where $W_q, W_k, W_v \in \mathbb{R}^{d \times d_h}$ are learnable weight matrices. Here, $F^w$ represents the transformed feature representation of the EEG window $W$. Temporal attention is computed as:

$$Z_t = \sum_{i=1}^{T} \text{softmax}\left(\frac{Q_t \cdot K_i}{\sqrt{d_k}}\right) \cdot V_i \tag{7}$$

where $d_k$ denotes the dimension of the Key vectors, used for scaling the attention scores.

### B. Spatial Attention

Inspired by the work of Zhao et al. [9], we introduced Spatial Attention to our model. Segmented windows $X \in \mathbb{R}^{W \times C}$ are extracted and constructed Pearson correlation [9] matrix $P \in \mathbb{R}^{C \times C}$ by the formula:

$$P_{ab} = \frac{\text{Cov}(W_a, W_b)}{\sigma W_a W_b} \tag{8}$$

where $W_a$ and $W_b$ are EEG signals of the $a$-th and $b$-th channels. Cov stands for the covariance, and $\sigma$ denotes the standardization operation. The correlation matrix then is used to compute Key (K) and Value (V) matrices applying the following formulas:

$$Q_{\text{spatial}} = W_q P \tag{9}$$

$$K_{\text{spatial}} = W_k P \tag{10}$$

where $W_q, W_v \in \mathbb{R}^{d \times d_h}$ are learnable weight matrices. As a result, the attention vector can be computed by the following formula:

$$Z_s = \sigma\left(\frac{Q_{\text{spatial}} \cdot K_{\text{spatial}}^T}{\sqrt{d}}\right) \tag{11}$$

where $\sigma(\cdot)$ represents the sigmoid activation function.

### C. Fusion Attention

Aiming at capturing the long-term dependencies, we introduce the Fusion Attention Mechanism, which allows the model to focus on crucial feature information in the whole EEG sequence, as in Fig. 2.

The $Q$ matrices are generated from the temporal feature, while the $K$ and $V$ matrices are derived by spatial feature. The QKV matrices can be computed using the following formulas:

$$Q_{\text{fusion}} = W_q Z_t \tag{11}$$

$$K_{\text{fusion}} = W_k Z_s \tag{12}$$

$$V_{\text{fusion}} = W_v Z_s \tag{13}$$

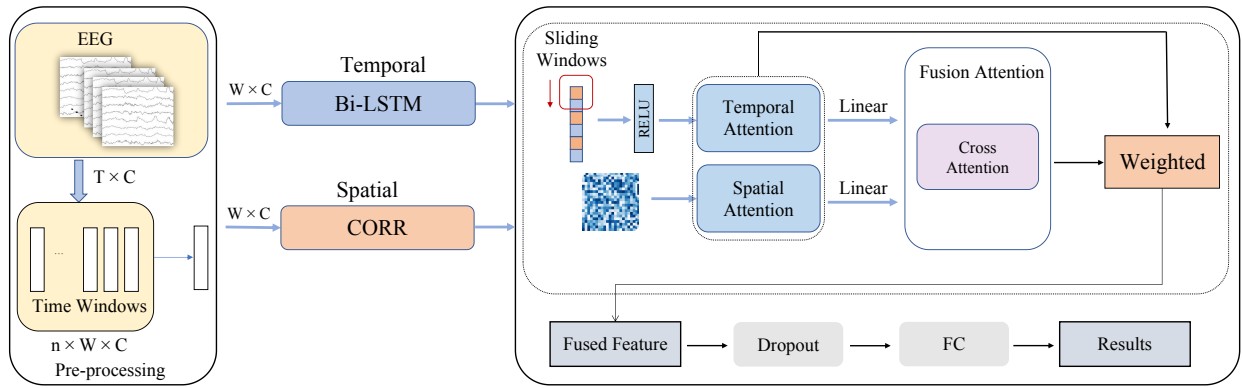

Fig. 1: The proposed method framework consists of three main modules: Pre-processing, Feature Extraction, and TSFA.

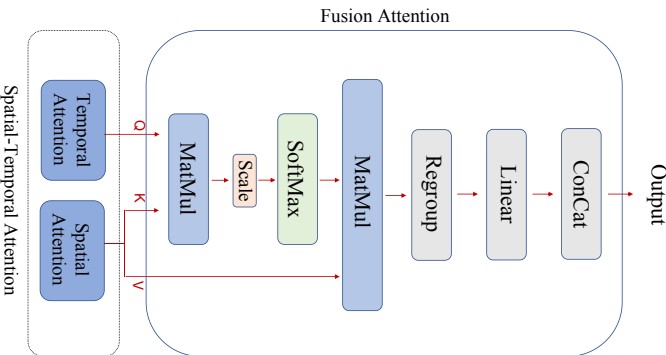

Fig. 2: The TSFA structure uses temporal features as Q and spatial features as K and V, integrating an attention mechanism to capture long-term dependencies in EEG signals.

where $W_q$, $W_k$, $W_v$ are learnable projection matrices. $Z_t$ represents the temporal attention, capturing the temporal dependencies within each window. $Z_s$ denotes the spatial attention input to the $K$ and $V$ matrices. We can derive the weighted sum by adopting the following formula:

$$Z_{\text{fusion}} = \text{Softmax}\left(\frac{Q_{\text{fusion}} \cdot K_{\text{fusion}}^T}{\sqrt{d}}\right) \cdot V_{\text{fusion}} \qquad (14)$$

### D. Evaluation Metrics

Epileptic seizures are classified as positive in the epilepsy detection task. We introduce accuracy, sensitivity, and F1 score to evaluate the performance of the model, using the following formulas:

$$\text{Accuracy} = \frac{\text{TP} + \text{TN}}{\text{TP} + \text{TN} + \text{FP} + \text{FN}} \qquad (16)$$

$$\text{Sensitivity} = \frac{\text{TP}}{\text{TP} + \text{FN}} \qquad (17)$$

$$\text{Precision} = \frac{\text{TP}}{\text{TP}} + \text{FP} \qquad (18)$$

$$\text{F1 Score} = 2 \cdot \frac{\text{Precision} \cdot \text{Sensitivity}}{\text{Precision} + \text{Sensitivity}} \qquad (19)$$

where TP (True Positive) stands for epileptic seizures being classified correctly and TN (True Negative) represents the non-seizure segments being classified as non-seizures. On the contrary, FP (False Positive) and FN (False Negative) represent the non-seizure segments misclassified as seizures and seizure segments that the model missed.

## III. EXPERIMENTS

The experiments used Python 3.12 and PyTorch 2.5.1 on an NVIDIA RTX 4090 GPU with 24 GB memory. Two evaluation strategies, 10-fold cross-validation and leave-one-patient-out cross-validation (LOPOCV), were employed to ensure model stability and generalization. For the 10-fold cross-validation, the dataset was split into 90% training and 10% testing sets for each fold, with 10% of the training set randomly allocated as a validation set. For LOPOCV, n-1 patients were used for training and one patient for testing, rotated across all patients, with 10% of the training set allocated as a validation set. Training involved a 0.001 learning rate, up to 50 epochs, the Adam optimizer, and early stopping with a 5-epoch patience if validation loss did not decrease.

### A. Dataset: CHB-MIT Dataset and Siena Scalp EEG Dataset

The CHB-MIT dataset [10] is widely used in training and testing seizure detection models. It contains EEG signals from Boston Children's Hospital included in the MIT EEG database, including EEG records from 24 patients, with data being seizures or non-seizures, and annotated seizure events. This dataset consists of scalp EEG recordings collected at a sampling rate of 256 Hz, with each recording containing multiple channels corresponding to different electrode placements based on the international 10-20 system [12]. 22 subjects contribute to the dataset (5 males, aged 3–22 years; 17 females, aged 1.5–19 years).

The Siena Scalp EEG Dataset [11] contains EEG recordings from 14 patients (9 males, aged 25–71; 5 females, aged 20–58) collected at the University of Siena's Unit of Neurology and Neurophysiology. Recorded at 512 Hz using a Video-EEG system with electrodes placed per the international 10–20 system, the dataset includes seizure and non-seizure events, annotated by expert clinicians based on International League Against Epilepsy criteria.

TABLE I: Performance Metrics for CHB-MIT and Siena Scalp EEG Datasets

| Strategy | Dataset | Accuracy (%) | Sensitivity (%) | F1 Score (%) |
|----------|---------|--------------|-----------------|--------------|
| 10-Fold | CHB-MIT | 97.15 | 97.06 | 96.45 |
| 10-Fold | Siena | 98.22 | 98.29 | 97.37 |
| LOPOCV | CHB-MIT | 92.86 | 92.95 | 91.72 |
| LOPOCV | Siena | 91.65 | 93.09 | 92.11 |

EEG data is segmented into 1-second slides, and non-seizure fragments were selected randomly to balance the amount of dataset between seizure and non-seizure samples, with non-seizure data being 2-4 times the amount of seizure data, enhancing the robustness and generality of the model.

### B. Data Pre-processing

Firstly, we use a 0.5–50Hz bandpass filter to remove possible noise or artifacts. Secondly, adopting ICA eliminates ocular and muscular artifacts. The data obtained is then normalized and standardized, and finally, it's segmented into 1s temporal windows for future feature extraction. The data pre-processing contributes to the enhancement of accuracy and robustness of the detection model.

## IV. RESULTS

### A. Overall Results

As shown in Table I, our model demonstrates strong performance, achieving a 10-fold cross-validation accuracy of 97.15% and an F1-score of 96.45% on the CHB-MIT dataset, and a 10-fold cross-validation accuracy of 98.22% with a leave-one-out cross-validation (LOPOCV) accuracy of 91.65% on the Siena dataset.

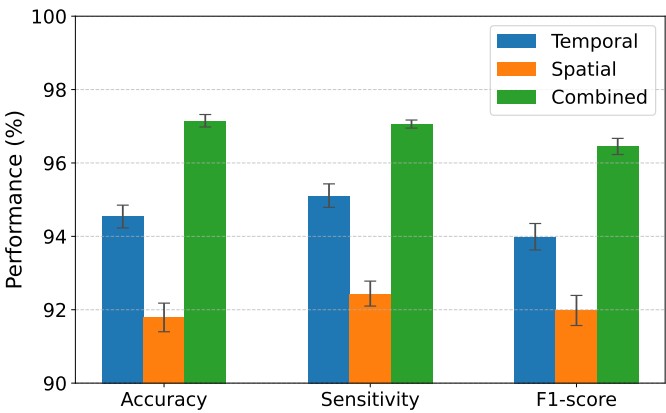

Fig. 3: Ablation experiment on individual attentions.

### B. Ablation Experiment

To evaluate the contribution of temporal attention, spatial attention, and overall performance, we conducted ablation experiments by systematically removing certain components and analyzing the impact of the model performance. The results of the ablation experiment are shown in Fig 3. The results demonstrate that both temporal and spatial attention contribute to the performance of the detection mode, but

temporal attention plays a more important role, as evidenced by its higher accuracy of 94.54%, sensitivity of 95.11%, and F1-score of 93.99%, compared to spatial attention's 91.79%, 92.44%, and 91.98%, respectively. This confirms that the temporal and spatial attention mechanisms complement each other, enabling the model to effectively capture both short-term dependencies within EEG sequences and spatial relationships across channels.

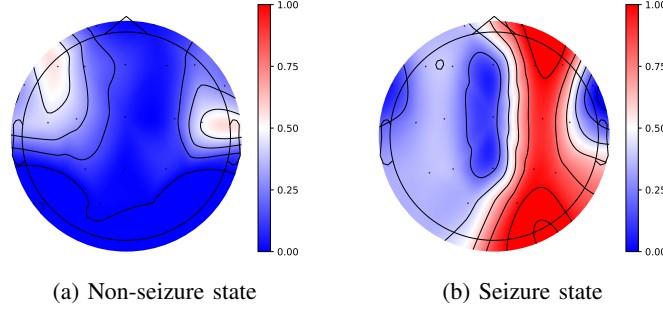

(a) Non-seizure state      (b) Seizure state

Fig. 4: Comparison of TSFA attention weights for patient chb 01 in non-seizure and seizure states via scalp topographic maps.

### C. The effectiveness of TSFA

To investigate the effectiveness of the TSFA module, we visualized the attention weights of TSFA for the seizure and non-seizure states of patient chb 01 using scalp topographic maps, as shown in Figure 4. The maps indicate that attention weights during a seizure state are significantly higher, with a pronounced concentration in the central scalp region, compared to the non-seizure state, where weights are predominantly lower with a more uniform distribution. This contrast demonstrates that our TSFA model can effectively capture the seizure regions and their associated features, distinguish the contributions of different channels based on the weight magnitudes, and accurately localize the seizure region. The elevated attention weights during seizure events, particularly in the central area, suggest that the model prioritizes channels likely linked to epileptic activity, further validated by the sharp gradient between high and low weight areas in the seizure state map.

### D. Comparison With Other Advanced Methods

We compared the performance of our TSFA model with existing deep learning models to verify TSFA's superiority. To ensure the fairness and transparency of the comparison, all methods followed the dataset and experimental settings selected in Section III. The comparison is conducted based on

TABLE II: Comparison with other methods on CHB-MIT dataset

| Authors | Year | Method | Accuracy (%) | Sensitivity (%) |
|---|---|---|---|---|
| Zhang et al. [13] | 2020 | AttVGGNet-RC | 95.28 | 95.07 |
| Shen et al. [6] | 2022 | DWT&RUSBoostedtree | 96.97 | 96.90 |
| Chen et al. [14] | 2024 | Spiking Conformer | 93.89 | 97.03 |
| Dokare et al. [15] | SHAP-RELFR | 97.10 | 95.67 | |
| Ours | 2025 | TSFA | 97.15 | 97.06 |

the CHB-MIT datasets (Table II). The results demonstrate that our model has a better performance than other existing models, which indicates the effectiveness of our model in epileptic detection. Compared with Shen et al., our model achieves the accuracy and sensitivity increase by 0.77% and 0.91%, respectively. These comparisons highlight the robustness and practical applicability of the TSFA model, making it a promising tool for accurate and reliable seizure detection in clinical environments.

## V. CONCLUSIONS

In this study, we develop the Temporal-Spatial Fusion Attention (TSFA) model to enhance seizure detection based on EEG signals by effectively capturing the fine-grained temporal dependencies and inter-channel relationships. The model integrates Temporal Attention and Spatial Attention for temporal and spatial features, which has an excellent performance verified by experiments comparing to existing models. Ablation experiments also show the superior effect of the combination of temporal and spatial attention. The strong robustness and generality demonstrated in the experiments indicate that it can be effectively applied to real-world epilepsy detection. The further study would focus on multimodal learning and adaptive fusion strategies, which will further enhance the robustness and generality, addressing challenges posed by heterogeneous data and improving clinical applicability.

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
