# OpenReview forum: "Feature Fusion Based on Temporal-Spatial Attention Model for Automatic Epileptic Seizure Detection"
_IEEE.org/EMBS/BHI/2025/Conference — BHI 2025_

### Official Review · Reviewer_EmpH · 2025-06-26
**High-Performing EEG Model for Seizure Detection, Needing Clearer Methodology and Fairer Comparisons**

**Confidence:** 4
**Clarity Of Writing:** good
**Clinical Significance:** good
**Methodological Novelty:** fair
**Overall Rating:** 3
**Final Rating:** 7

**Experiments And Results:**

great

**Questions For The Authors:**

To better assess the significance and reproducibility of your results, we would appreciate more details on the experimental methodology:

- What learning rate and number of training epochs were used?
- Which optimizer was selected? Were techniques such as learning rate scheduling or early stopping employed?

Secondly, we would welcome a clarification on how your work compares to prior state-of-the-art methods, particularly given that some referenced papers perform different classification tasks. Could you explain why these comparisons remain meaningful despite the differences in problem formulation?

A clear and thorough response to these questions — along with revisions that address the minor weaknesses noted — could strengthen the paper and potentially warrant an evaluation of "borderline accept" or "weak accept."

**Strengths:**

Here are the strengths of the paper, listed in order of importance:

- $\text{\textbf{Strong model performance and well-justified architecture}}$: The proposed Temporal-Spatial Fusion Attention (TSFA) model achieves 97.15% accuracy on the CHB-MIT dataset in classifying ictal vs. non-ictal EEG data. The architecture’s use of temporal and spatial attention, followed by a fusion attention mechanism, appears to contribute to its effectiveness. Additionally, although the paper does not focus on interpretability, the use of attention mechanisms opens the door to post-hoc interpretation tools such as attention rollout [1].
- $\text{\textbf{Detailed architecture presentation}}$: Sections 2.A–2.C clearly describe each component of the model (temporal, spatial, and fusion attention), supported by formulas (1-14) and visual diagrams (Figs. 2 and 3). This helps readers follow the design logic and understand how the system processes EEG signals.
- $\text{\textbf{Clinical relevance}}$: The paper addresses an important health challenge. Improving automatic seizure detection has potential real-world impact, such as supporting clinicians in diagnosis or enabling early warning systems to improve quality of life for epilepsy patients.

[1] Abnar, Samira, and Willem Zuidema. "Quantifying attention flow in transformers." arXiv preprint arXiv:2005.00928 (2020).

**Summary Of The Paper:**

This paper proposes the Temporal-Spatial Fusion Attention (TSFA) model to classify EEG data from epileptic patients as ictal (seizure) or non-ictal. The model uses a Bi-LSTM to extract temporal features and computes Pearson correlation to capture spatial dependencies between EEG channels. These features are fused via an attention mechanism to enhance detection.

The model is evaluated on the CHB-MIT dataset using 10-fold cross-validation, achieving 97.15% accuracy and 97.06% sensitivity, outperforming several recent methods on the same dataset. An ablation study highlights the importance of combining both temporal and spatial features, with each contributing to performance gains, though temporal attention has slightly more impact.

**Weaknesses:**

Weaknesses of the paper have been separated between major and minor, and listed by order of importance.

Major Weaknesses:
- $\text{\textbf{Lack of background and insuficient referencing}}$: While the comparative evaluation with recent works is a strength, the paper lacks sufficient theoretical and contextual grounding. The introduction offers minimal discussion of related literature, despite seizure detection from EEG being a highly active research area. Core components of the proposed method — such as the use of Bi-LSTM, Pearson correlation for spatial features, and the 10–20 EEG system — are introduced without adequate citation. Similarly, foundational clinical explanations of epilepsy are presented without supporting references.
- $\text{\textbf{Inconsistent benchmarking due to differing tasks}}$: The paper claims state-of-the-art performance by outperforming recent methods [5, 8–11] on the CHB-MIT dataset. However, several of the referenced works address different classification tasks. For example, [5] focuses on distinguishing interictal and preictal states, while [10] performs a three-class classification (interictal, preictal, ictal). These tasks differ significantly from binary ictal/non-ictal classification and therefore undermine the validity of direct performance comparisons. The paper's claim of superiority is thus overstated unless the results are re-evaluated on a common, well-defined task.
- $\text{\textbf{Incomplete experimental methodology}}$: The paper lacks key details necessary for reproducibility. Important training parameters such as learning rate, number of epochs, optimizer choice, and use (or absence) of techniques like learning rate scheduling or early stopping are not reported. This omission limits the ability of other researchers to replicate or build upon the work.
- $\text{\textbf{Unsubstantiated claim of robustness against class imbalance}}$: The paper claims robustness to imbalanced data in the introduction. However, the experimental section reveals that non-seizure samples were randomly downsampled to balance the dataset. Moreover, no experiments were performed on naturally imbalanced settings to validate this claim. This inconsistency weakens the credibility of the robustness assertion.

Minor weaknesses:
- Given the use of 10-fold cross-validation, reporting standard deviations for accuracy, sensitivity, and F1-score would help assess the stability of the model and the statistical significance of the reported improvements.
- Some content is unnecessarily repeated. For example, Section III-B reiterates information already mentioned earlier, and the performance results in Section IV-B repeat exact performance values stated in prior sections.
- Variables $F^w$ and $d_k$ are not defined in section II.
- Figure 1 is informative but could benefit from larger labels and clearer layout, especially on the left-hand side.
- Reference [4] does not appear to be cited in the text.
- Equation 1 and 2 contain a typo, having each a trailing "#(1)" or "#(2)". Similarly, the equation of precision is incorrect.
- After Equation (8), the text incorrectly states that “Cov” stands for “correlation”; it should correctly refer to covariance.

---

### Official Review · Reviewer_hM6w · 2025-06-27
**Good modeling with weak experiment results**

**Confidence:** 4
**Clarity Of Writing:** great
**Clinical Significance:** fair
**Methodological Novelty:** good
**Overall Rating:** 6
**Final Rating:** 7

**Experiments And Results:**

fair

**Questions For The Authors:**

How are the EEG signals from each patient used for training and evaluation? Is the data from a single patient used in both training and evaluation?

What is the procedure to compare the proposed model with existing models? Are the existing models trained and evaluated in the same settings as the proposed model?

**Strengths:**

The paper introduces several attention modules to extract temporal and spatial information from the EEG signals to detect epilepsy.

**Summary Of The Paper:**

The paper developed a deep learning model for automatic epilepsy detection using EEG signals as inputs. The model features three different self-attention modules to extract useful information from different aspects of the input signals. The experiment results suggest that the proposed model offers improved performance and robustness compared to existing deep learning solutions for the same task.

**Weaknesses:**

The experiment and comparison setup is not well-explained, undermining the credibility of the claims made by the authors.

---

### Official Review · Reviewer_yHFP · 2025-07-12
**Feature Fusion Based on Temporal-Spatial Attention Model for Automatic Epileptic Seizure Detection**

**Confidence:** 5
**Clarity Of Writing:** good
**Clinical Significance:** good
**Methodological Novelty:** good
**Overall Rating:** 7

**Experiments And Results:**

good

**Questions For The Authors:**

Considering that your evaluation is based solely on the CHB-MIT dataset, have you considered testing your model on additional EEG datasets to evaluate its generalizability across different patient populations and recording settings?

While attention mechanisms are often linked to interpretability, could you clarify whether the attention weights in your model correspond to clinically meaningful EEG regions, such as seizure onset zones identified by neurologists?

**Strengths:**

The paper presents several strengths that make the proposed TSFA model highly effective for epileptic seizure detection. The integration of temporal and spatial attention mechanisms allows the model to capture both dynamic EEG signal patterns over time and spatial relationships across different brain regions, which improves overall accuracy and robustness. The use of Bi-LSTM enhances the model’s ability to process sequential data, while Pearson correlation effectively models inter-channel dependencies. The fusion attention mechanism enables the model to integrate information from both domains, leading to more discriminative feature representations. Experimental results on the CHB-MIT dataset demonstrate strong performance, outperforming several recent models in accuracy, sensitivity, and F1-score. Ablation studies further confirm that the combined use of temporal and spatial attention contributes significantly to the model’s success, showing the benefit of a synergistic approach. The paper also emphasizes real-world applicability by addressing challenges such as data imbalance and noise, supporting the model’s potential for clinical use.

**Summary Of The Paper:**

The paper presents the Temporal-Spatial Fusion Attention (TSFA) model for automatic epileptic seizure detection using EEG signals. It addresses challenges in EEG-based seizure detection, such as noise, data imbalance, and the need to capture both temporal and spatial patterns. The proposed model combines Bi-LSTM for extracting temporal features and Pearson correlation for modeling spatial relationships across EEG channels. These features are integrated using an attention-based fusion mechanism to enhance classification performance. Evaluated on the CHB-MIT dataset, the TSFA model achieves high accuracy, sensitivity, and F1-score, outperforming several state-of-the-art approaches. Ablation studies confirm the complementary roles of temporal and spatial attention, with the full fusion model yielding the best results. The authors conclude that their model is robust, generalizable, and promising for clinical seizure detection, and propose future work in multimodal learning and adaptive fusion strategies to further improve performance.

**Weaknesses:**

The paper presents a highly accurate and well-structured model for epileptic seizure detection using EEG signals, but a few very minor weaknesses can be noted. The evaluation is limited to a single dataset, CHB-MIT, which may restrict insights into the model's generalizability across diverse patient populations or recording environments. Additionally, while the model’s architecture is sophisticated, the interpretability of attention weights in clinical terms is not deeply explored, which could limit immediate clinical adoption or understanding by non-technical stakeholders.